# SV40 Transfected Human Anterior Cruciate Ligament Derived Ligamentocytes—Suitable as a Human in Vitro Model for Ligament Reconstruction?

**DOI:** 10.3390/ijms21020593

**Published:** 2020-01-16

**Authors:** Gundula Schulze-Tanzil, Philipp Arnold, Clemens Gögele, Judith Hahn, Annette Breier, Michael Meyer, Benjamin Kohl, Michaela Schröpfer, Silke Schwarz

**Affiliations:** 1Institute of Anatomy and Cell Biology, Paracelsus Medical Private University, Salzburg and Nuremberg, 90419 Nuremberg, Germany; clemens.goegele@pmu.ac.at (C.G.); schwarzbiggy@yahoo.de (S.S.); 2Institute of Anatomy, Christian Albrecht Universität, Otto-Hahn-Platz 8, 24118 Kiel, Germany; p.arnold@anat.uni-kiel.de; 3Department of Biosciences, Paris Lodron University Salzburg, 5020 Salzburg, Austria; 4Leibniz-Institut für Polymerforschung Dresden e. V., 01069 Dresden, Germany; hahn-judith@ipfdd.de (J.H.); breier@ipfdd.de (A.B.); 5Forschungsinstitut für Leder und Kunststoffbahnen (FILK) gGmbH, 09599 Freiberg, Germany; michael.meyer@filkfreiberg.de (M.M.); michaela.schroepfer@filkfreiberg.de (M.S.); 6Charité – Universitätsmedizin Berlin, corporate member of Freie Universität Berlin, Humboldt-Universität zu Berlin and Berlin Institute of Health, Department of Traumatology and Reconstructive Surgery, Campus Benjamin Franklin, Hindenburgdamm 30, 12203 Berlin, Germany; benjamin.kohl@charite.de

**Keywords:** anterior cruciate ligament, tissue engineering, PLA, P(LA-CL), embroidered scaffold, ligamentocytes, SV40

## Abstract

Cultured human primary cells have a limited lifespan undergoing dedifferentiation or senescence. Anterior cruciate ligaments (ACL) are hypocellular but tissue engineering (TE) requires high cell numbers. Simian virus (SV) 40 tumor (T) antigen expression could extend the lifespan of cells. This study aimed to identify cellular changes induced by SV40 expression in human ACL ligamentocytes by comparing them with non-transfected ligamentocytes and tissue of the same donor to assess their applicability as TE model. Human ACL ligamentocytes (40-year-old female donor after ACL rupture) were either transfected with a SV40 plasmid or remained non-transfected (control) before monitored for SV40 expression, survival, and DNA content. Protein expression of cultured ligamentocytes was compared with the donor tissue. Ligamentocyte spheroids were seeded on scaffolds embroidered either from polylactic acid (PLA) threads solely or combined PLA and poly (*L*-lactide-co-ε-caprolactone) (P(LA-CL)) threads. These scaffolds were further functionalized with fluorination and fibrillated collagen foam. Cell distribution and survival were monitored for up to five weeks. The transfected cells expressed the SV40 antigen throughout the entire observation time, but often exhibited random and incomplete cell divisions with significantly more dying cells, significantly more DNA and more numerous nucleoli than controls. The expression profile of non-transfected and SV40-positive ligamentocytes was similar. In contrast to controls, SV40-positive cells formed larger spheroids, produced less vimentin and focal adhesions and died on the scaffolds after 21 d. Functionalized scaffolds supported human ligamentocyte growth. SV40 antigen expressing ligamentocytes share many properties with their non-transfected counterparts suggesting them as a model, however, applicability for TE is limited.

## 1. Introduction

Cultured primary cells have a restricted lifespan and often display an instable phenotype undergoing dedifferentiation. Tissue engineering (TE) approaches require high cell numbers with a differentiated phenotype, but human-derived cells are only available in limited numbers and are harvested in many cases from elderly and individuals with background diseases. Unfortunately, animal-derived cells usually show inter-species differences strongly limiting transferability of results. To utilize human cells, they can be immortalized and show a delayed or no senescence thereafter.

Many types of human cells cultured in vitro can be transfected with the simian virus (SV) 40, a DNA tumor virus [1]. Expression of the SV40 large tumor (T) antigen mediated by the virus has been used for immortalization of human periodontal ligamentocytes [1,2,3,4]. The authors of these previous studies reported an increase in lifespan and a maintenance of intrinsic cell characteristics. They concluded that the immortalized cells represent a useful tool to study biology and regenerative mechanisms of this cell type [2,3,4]. In another study a human cementifying fibroma cell line was created with a temperature-sensitive SV40 plasmid [1]. However, an immortalized anterior cruciate ligament (ACL) cell line has not been reported so far.

Kudo et al., observed that the cells transfected with the SV40 T antigen exhibited a “crisis” state between passages 22 and 35, but additional transfection with human telomerase reverse transcriptase (hTERT) in the SV40-transformed human cardiac fibroblasts provided a strategy to bypass the crisis and allow cell maintenance over 200 passages [1]. In contrast to the above mentioned studies, where no changes in cell characteristics due to SV40 transfection were detected McNutt et al., reported that some cytoskeletal filaments revealed an altered distribution in immortalized 3T3 cells. They were reduced in the anterior extensions of transformed fibroblasts, containing large blunt pseudopodia and relatively few microvilli [5]. Salingcarnboriboon et al., isolated an Achilles tendon-derived stem cell-like cell line of murine origin containing a temperature-sensitive mutant of the SV40 T antigen, which displayed expression of tendon-specific markers such as scleraxis [6]. However, Salingcarnboriboon et al., did not find any differences between SV40 T antigen expressing and SV40 negative murine tendon-derived cells. So far, there are no published studies reporting transfection of ACL-derived fibroblasts with the SV40 T antigen and application for TE. The ACL contains only few ligamentocytes representing highly specialized fibroblasts embedded into an abundant extracellular matrix (ECM), which consists mainly of type I collagen fibers arranged in parallel bundles. Tendon- and ligament-derived cells including ligamentocytes from the ACL are characterized by expression of the transcription factor scleraxis [7], Mohawk [8], the fibroblast/ligament marker tenascin C, the mesenchymal marker vimentin and diverse less specific ECM components such as type III collagen, fibronectin as well as decorin, by a weak synthesis of aggrecan and other components [9,10,11].

Therefore, the present study aimed to validate crucial properties of SV40 transfected ACL ligamentocytes in direct comparison with untreated cells and ACL tissue from the identical donor to assess their applicability as a human cell-based TE model. In addition, it was examined whether human ACL-derived ligamentocytes colonize a novel scaffold composition embroidered from polylactid acid (PLA) threads or PLA-combined with poly(*L*-lactide-co-ε-caprolactone) (P(LA-CL)) threads and functionalized by fluorination and collagen foams.

## 2. Results

### 2.1. ACL Donor Tissue Characterization

Hematoxylin eosin (HE) staining revealed the typical ACL tissue structure: few cells were detectable between ECM fiber bundles. Alcian blue (AB) staining depicted only a faint staining of sulphated glycosaminoglycans (sGAGs). Elastic fibers were barely detectable (Appendix A). An abundant immunoreactivity for types I and III collagens, presence of decorin the most abundant proteoglycan in ligaments, aggrecan, and glycoproteins such as fibronectin (Appendix A) as well as the fibroblast/ligamentocyte marker tenascin C (Figure 1) could be demonstrated in the donor tissue. However, lubricin and type II collagen were only faintly visible (Figure 1). The hyaluronan receptor (CD44) as a membrane component of many mesenchymal cells was abundantly expressed in the ligamentocytes embedded within the tissue (Figure 1). In regard to cytoskeletal proteins a clear F-actin staining became evident. The degeneration score value according to the classification used by Hasegawa et al., was 3.5, which describes only a mild degeneration of the donor tissue [12].

### 2.2. SV40 T Antigen Expression

In the isolated and transfected ligamentocytes SV40 T antigen protein expression could be proven (Figure 2). The majority of transfected cells was SV40 positive, whereas non-transfected ligamentocytes and the donor tissue samples were negative. Transfected ligamentocytes displayed a clear nuclear signal. F-actin staining did not differ between transfected and non-transfected ligamentocytes and also the ligamentocytes in the native donor tissue showed the presence of dense F-actin filaments (Figure 2). ACL ligamentocytes within the tissue depicted a typical fibroblastic morphology, but were generally smaller, with smaller cell nuclei than those cultured in monolayer.

### 2.3. Morphology and Cell Survival of Non-Transfected and SV40 Transfected ACL Ligamentocytes in Monolayer Culture

ACL ligamentocytes migrating out of the explant tissue formed monolayers, which were expanded and used for SV40 transfection. Microscopic analysis revealed that non-transfected (control) and transfected ligamentocytes differed morphologically. The SV40 transfected cells had a more compact shape of the cell bodies compared to the control cells which were more slender (Figure 3). SV40 transfected ligamentocytes showed morphological stability and maintained proliferation for more than 20 passages. Non-transfected ligamentocytes changed their morphology in passage 9 forming clusters of round and proliferating cells (Figure 3A3). In contrast to the non-transfected ligamentocytes of the same donor, the counted cell number of the SV40 transfected cells was higher, but significantly more cells died in monolayer culture as detectable by Live/Dead assay at 48 h (Figure 3).

### 2.4. Cell Nuclei and Numbers of Nucleoli in Non-Transfected and SV40 Transfected ACL Ligamentocytes in Monolayer Culture

In contrast to non-transfected control ligamentocytes, SV40 transfected cells had cell nuclei with larger diameters (Figure 4). Cell nuclei were often polymorphic. In addition, transfected cells underwent chaotic and incomplete cell divisions (e.g., no or delayed cytokinesis) and contained more nucleoli (control: 1.8 ± 2.1 per cell nucleus and SV40 transfected: 4.9 ± 0.9/ cell nucleus), visible in HE stained cells after 48 h of culture (Figure 4).

### 2.5. DNA Content in Monolayer Culture in Non-Transfected and SV40 Transfected ACL Ligamentocytes in Monolayer culture

The DNA content was measured after 24, 48, and 72 h of monolayer culture to estimate DNA synthesis. Compared to the non-transfected control cells, the DNA content in SV40 transfected ligamentocytes was significantly higher at all points in time investigated (Figure 5).

### 2.6. Protein Expression Analysis in Non-Transfected and SV40 Transfected Ligamentocytes in Monolayer Culture

Typical ligament ECM components such as type I collagen, fibronectin, decorin, aggrecan and lubricin were expressed by both monolayer-cultured ligamentocyte populations. The fibroblast/ligamentocyte marker tenascin C (Figure 6) was detectable in transfected and non-transfected ligamentocytes and also the tenocyte/ligamentocyte marker Mohawk (Appendix A). However, some cytoskeletal components such as vinculin-associated focal adhesions and vimentin were reduced in SV40 transfected compared to the control cells, whereas other proteins did not differ including the hyaluronan receptor CD44, the cell-matrix receptor β1-integrin, focal adhesion kinase (FAK), talin and the myofibroblast marker α-Smooth Muscle Actin (αSMA) (Figure 6).

### 2.7. Gene Expression of Typical Tendon Components in Non-Transfected and SV40 Transfected Ligamentocytes

Gene expression of the main ligament ECM protein collagen type I and the ligament marker scleraxis could be shown in both, non-transfected and SV40 transfected cells. There was no significant difference in collagen type I and scleraxis gene expression between non-transfected cells and the cells bearing the SV40 antigen (Figure 7).

### 2.8. Survival of Non-Transfected and SV40 Transfected Ligamentocytes in 3D Culture

By using the hanging drop method (50,000 and 150,000 cells per spheroid) SV40 transfected cells produced only instable spheroids compared to the control ligamentocytes of the same donor. In addition, SV40 transfected cells died on the scaffolds after 21–28 d independent of the spheroid size of 50,000 [Figure 8 and Figure 9] or of 150,000 [not shown] cells per spheroid in 3D scaffold culture. The non-transfected cells, however, survived until the end of the observation period of five weeks.

There was no major difference in the colonization of P(LA-CL)/PLA and PLA scaffolds by non-transfected and SV40 transfected ACL ligamentocytes detectable. However, on both scaffold variants transfected cells demonstrated a more compact morphology in comparison to the slender non-transfected ligamentocytes (Figure 8 and Figure 9).

Non-transfected and transfected cells were seeded in spherical plates to form spheroids and to compare the cell behavior in tiny spheroids. Spheroids of SV40 transfected cells were significantly larger compared to non-transfected cells irrespectively of initial cell numbers used for the spheroid formation (Figure 10). The amount of dead cells was higher in spheroids produced by SV40 transfected cells compared with those consisting of non-transfected ligamentocytes (Figure 10). The size of spheroids was cell number-dependent e.g., 70.54 ± 11.94 µm vs. 120.27 ± 18.09 µm for 200 ACL ligamentocytes per spheroid (non-transfected vs. SV40 transfected) or 117.41 ± 21.95 µm vs. 167.56 ± 25.42 µm for 1000 cells per spheroid (non-transfected vs. SV40 transfected).

## 3. Discussion

Human derived cells represent a more valuable in vitro model than animal-derived cells due to interspecies differences. However, availability of human healthy tissue is highly limited, particularly of those tissues with very low cell content such as ligaments. TE experiments require large amounts of cells. Hence, an immortalized and stable cell line based on human ligamentocytes would be of high advantage for musculoskeletal research. The expression of the SV40 large T antigen, a multifunctional regulatory protein, is known to induce a delay of senescence in human cells, as shown for periodontal ligamentocytes for approximately 30–40 passages [4,13]. Human cells are generally semi permissive for replication of the SV40 antigen [14,15]. The SV40 transgene is replicated extrachromosomally during the synthesis phase of cell cycle. The stimulated DNA synthesis, observed in SV40 T antigen transfected human fibroblasts seems to be the response to binding and inactivation of p53 and the retinoblastoma tumor suppressor genes normally causing growth arrest in quiescent and senescent cells by regulating cell cycle proteins [16,17]. However, no tumor formation could be shown when periodontal ligamentocytes transduced with a viral vector mediating SV40 T antigen expression were introduced into a BALBc nude mice model [13]. The expression of SV40 T antigen in primary human ACL ligamentocytes could be shown in the present study.

Distinct morphological changes of cultured cells in response to SV40 transformation have previously been described investigating the periodontal ligament cell line already mentioned above [13]. As described in the study of Parkar et al., no difference in cell size could be detected using flow cytometry, but a more granular cell structure of periodontal cells was observed [4]. For flow cytometry it has to be considered that cells have to be detached and suspended cells generally exhibit a rounded cell shape. Therefore, the unique pattern of shape, spreading and formation of filo- and lamello-podia of adhering cells cannot be determined by this method.

SV40 T antigen expression could be proven in transfected cells with a clear nuclear signal during the entire culture period as normally expected [18]. Compared to non-transfected ligamentocytes transfected cells showed morphological differences such as a more compact cell shape and significantly larger diameters of cell nuclei. Therefore, the cytoskeleton was examined to find an explanation for the observed morphological changes. Vimentin as cytoskeletal intermediate filament, which stabilizes cell shape was more or less confined to the cell body and did not extend into the cellular extensions in the SV40 transfected ACL ligamentocytes compared to the non-transfected controls. A higher mitotic activity could contribute to delayed intermediate filament reorganization. The research group of Hoang et al., also found an enhanced proliferative activity in SV40 transformed periodontal cells [13]. The DNA content measured in the present study was indeed significantly higher than in the non-transfected controls suggesting a higher proliferative activity. However, the larger cell nuclei observed in transfected compared to non-transfected cells could be attributed to an increased DNA content, too. Nevertheless, calculations of total cell numbers of SV40 expressing ligamentocytes confirmed higher cell amounts in the transfected population compared with the non-transfected control (Figure 4). In addition, the significantly larger spheroid sizes with transfected cells suggested a more rapid proliferation. The inhomogeneous cell divisions possibly caused by chromosomal aberrations might be responsible for higher rates of dead cells and cell fragments in the SV40 transfected cell population in monolayer, spheroid and scaffold cultures. Characteristic chromosomal patterns reflecting imbalances in response to SV40 antigen expression have been demonstrated previously in human fibroblasts [19]. Some chromosomes or segments of them were present in excess, others were lost. Accordingly, several metabolic imbalances could also be detected in the same study [19].

Nucleoli are defined areas within the cell nuclei, where ribosomal rRNA is produced indicating activity of protein synthesis. Hence, it can be suspected that SV40 transfected cells exert a higher protein synthesis. It could be a direct result of the induced production of the SV40 T antigen. Nevertheless, the gene expression of the most abundant ECM protein in tendon and ligaments, type I collagen, did not significantly differ between the SV40 transfected and non-transfected cells. Both cell populations expressed the main ligament marker scleraxis and Mohawk (Appendix A) proving their differentiated phenotype. The main components of ligaments including type I and III collagens, tenascin C, decorin, aggrecan and only minor amounts of the cartilage collagen type II and lubricin could be demonstrated in the donor tissue sharing close similarities with the expression profile in both monolayer cultured cell populations.

This suggests that both, the non-transfected and the SV40 transfected population, can be used as an in vitro model and for ligament TE.

Cytoskeletal changes might be responsible for impaired cell adhesion and survival in 3D scaffold culture. However, it remains unclear why explicitly vimentin, a mesenchymal cytoskeleton intermediate filament, which ensures cell stability, showed a reduced distribution within the cells and may be responsible for their morphological differences. Vimentin organization is known to change substantially during mitosis and early cell adhesion, forming transient ring- and ball-like structures [20]. Therefore, the reduced vimentin immunoreactivity could result from accelerated and disorganized cycles of cell division in the transfected cells. The formation of focal adhesion sites positive for cytoskeletal vinculin also changed, but not the expression of other selected cytoskeletal proteins such as talin focal adhesion kinase (FAK) and the important cell-ECM receptor β1-integrin. Talin and FAK are involved in integrin activation and integrin-mediated cell adhesion [21]. The inability to form stable larger spheroids observed with the SV40 expressing ligamentocytes might also be a result of reduced cell-cell adhesion. It has been reported that particularly fibronectin is responsible for spheroid formation [22,23] but fibronectin immunoreactivity did not differ between both cell populations (Figure 3). However, small spheroids consisting of 200 or 1000 cells could be produced more stably than larger spheroids (50,000; 150,000 cells per spheroid) with both cell populations. Nevertheless, a higher rate of dead cells could be detected in the transfected population. This suggests that not only the higher rate of cell death due to chromosomal defects but also limited diffusion through the larger spheroid as a basis for nutrition might play a crucial role here. To prove this in detail, also cadherins, known as typical cell–cell-adhesion molecules could be studied in future. The scaffold type selected here for 3D culture has been proven in several previous studies to be highly cytocompatible using a lapine primary ACL cell model [24,25,26]. The scaffolds were functionalized by fluorination and supplemented with collagen foam to provide multiple motifs for cell adhesion. However, the lapine ACL cell system used in all previous studies [24,25,26] might differ from human ACL ligamentocytes. Therefore, as a result of our present study we can conclude that scaffold composition is suitable for TE with human-derived ligamentocytes. Spheroids have previously been proven to be a promising strategy for directed seeding of scaffolds with ligamentocytes [11,24,27].

The fact that the main expression profile for typical ligament proteins did not change and the cells acquired an extended lifespan makes the SV40 T antigen-positive ACL cells attractive as a model system. The cells could be cryopreserved and recovered for further culturing. Primary cells such as tenocytes which strongly resemble ligamentocytes, generally show a phenotypic drift during passaging already between the passages 1–8 [28]. We found a dedifferentiation of the non-transfected cells used in the present study at passage 9 (Figure 4). Cells from degenerated tissue could also reflect a phenotypic shift as reported previously [29,30]. Hence, the degeneration score of the donor tissue was assessed here, where we only found a mild degeneration, possibly the precondition for the ACL rupture observed in the donor patient. Normally, it is difficult to get unimpaired human ligament tissue samples for research.

In this pilot study, a transient transfection strategy was selected to avoid higher safety requirements. The transfection was repeated several times for the maintenance of the transgene, which could be nicely proven (Figure 2). The telomeres are still shortened since non-tumorigenic mature somatic cells have no telomerase activity during the lifespan extension mediated by the SV40 T antigen. Hence, the cell population can enter into a state referred to as “crisis“, leading to cell death [1]. In future studies a retroviral vector could be used for permanent SV40 transformation which allows integration of the transgene into host chromosomes [4,13]. In addition to the SV40 antigen transfection, hTERT transfection would help in future to further delay senescence. It could restore telomerase activity and prevent telomeric shortening during mitosis, which is associated with senescence in somatic cells [1,2,31].

## 4. Materials and Methods

### 4.1. Cell isolation by Explant Culture

ACL samples (one female donor, age 40), were harvested in accordance with the institutional ethical committee of the Charité-Universitätsmedizin Berlin, Campus Benjamin Franklin (ethical approval number EA4-033-08, approval date: 22 May 2008). Surrounding connective tissue, including the synovial membranes, was removed. The pure ligament tissue was cut into 1–2 mm^2^ slices and incubated in T25 cell culture flasks (Sarstedt, Nümbrecht, Germany) with growth medium (Ham’s F-12/Dulbecco’s Modified Eagle’s [DMEM] Medium 1:1) containing 10% fetal calf serum (FCS), 10,000 IU/mL penicillin/10,000 µg/mL streptomycin, 2.5 µg/mL amphotericin B, non-essential amino acids (all from Biochrom, Berlin, Germany) and 25 µg/mL ascorbic acid (Sigma-Aldrich, Munich, Germany) at 37 °C and 5% CO_2_. After 1–2 weeks, the ACL ligamentocytes started to migrate from the tissue slices. Subsequently, ligamentocytes were harvested using 0.05% trypsin/0.02% EDTA (Biochrom) and expanded in T75 and T175 culture flasks. For analyzing the expression of marker protein profile on protein level, ACL ligamentocytes were seeded on poly-L-lysin (Biochrom) coated cover slips at 1 × 10^4^ cells/cm².

### 4.2. Transfection with an SV40 Plasmid

Human ACL ligamentocytes (female donor, 40 years old) were SV40 transfected in three repetitive cycles using 2 × 10^6^ cells starting at passage five. Each cycle contained a transient transfection using Fugene (3 µl stock solution/1 µg DNA plasmid; Qiagen, Hilden, Germany) with a simian virus 40 T antigen (SV40) containing plasmid (5 µg), cloned into a pc.DNA3.1+ backbone. After transfection with Fugene medium was exchanged after 4 h to minimize cell-toxic effects. After 48 h cells were splitted to 60% confluence and the transfection was repeated. After the last round of transfection cells were passaged for 10 times to allow SV40 positive cells (spontaneous integration into the genome) to overgrow SV40 negative ones. Positivity for SV40 was evaluated using immunofluorescence microscopy against the SV40 antigen.

### 4.3. Scaffold Preparation

Scaffolds with a dimension of 28 mm length, 4 mm width, and 2 mm thickness were embroidered using either solely PLA threads (Tt = 155 dtex, pellets from NatureWorks LLC, Minnetonka, MN, USA), which were melt spun to a multifilament consisting of six filaments at the IPF (Dresden, Germany) or PLA threads combined with P(LA-CL) threads (monofilament suture thread in USP 7-0, Gunze Ltd., Osaka, Japan). Scaffold preparation was performed using an embroidery machine as reported previously (Type: JCZ 0209-550, ZSK Stickmaschinen GmbH, Germany) [32,33]. Scaffolds had a zig-zag stitch pattern with stitch length 1.8 mm, stitch angle 15°, duplication shift 0.2 mm and consisted of three plies which were locked together. The scaffolds were prepared on a water-soluble fabric made of polyvinyl alcohol (PVA, Freudenberg Einlagestoffe KG, Weinheim, Germany) that was removed by thorough washing out three times for each step with 30 min in distilled water on a compact shaker (Type: KS 15 A, Edmund Bühler GmbH, Bodelshausen, Germany). Then, scaffolds were dried at room temperature (RT).

For fluorination the scaffolds were incubated in a fluorination batch reactor (Fluor-Technik-System GmbH, Lauterbach, Germany) in a mixture of 10% fluorine gas in synthetic air for 60 s. Scaffolds were supplemented with lyophilized re-fibrillated collagen foam (prepared from purified native bovine collagen type I) which was cross-linked for stabilization with hexamethylene diisocyanate (HMDI). Cross-linking was performed with the gas-phase of HMDI in an exsiccator. Finally, the samples were sterilized by incubation in 70% ethanol for 30 min followed by rinsing gently three-times in A. dest. before testing in cell culture.

### 4.4. Spheroid Culture and Scaffold Seeding

Spheroid formation was achieved using the hanging drop method (50,000 cells per drop), pellet culture (150,000 cells per pellet sedimented in an Eppendorf tube by centrifugation for 5 min at 500 *g* and cultivation for 3 d) or culturing 200–1000 cells per spheroid using a 24 well spherical Kugelmeier 5D plate (Kugelmeier, Zurich, Switzerland). For the hanging drop method, a cell suspension (50,000 cells per 50 µL) was dropped on the lid of a non-coated petri dish (nerbe plus, Winsen/Luhe, Germany). The lid was turned back on the bottom of the dish and cultured for 72 h until spheroids were ready to be harvested. For scaffold seeding, 50 spheroids carrying 50,000 cells or 15 pellets containing 150,000 cells each were placed on PLA or P(LA-CL)/PLA scaffolds. The spheroids were allowed to adhere to the scaffolds for 48 h in a static culture before being transferred to a 50 mL Tubespin^®^ Bioreactor tube (TPP^®^, Trasadingen, Switzerland) and dynamically cultivated for 7–35 d. In addition, to get deeper insights into spheroid formation of non-transfected and transfected cells very small spheroids were produced based on different smaller cell numbers (200 and 1000 cells) using a 24 well spherical plate 5D. Experiments were performed in independent triplicates. Cells were aggregated for spheroid formation for 5 d with a growth medium exchange after 24 h.

### 4.5. Culture of ACL Ligamentocytes on Glass Cover Slips

For analyzing the expression profile of marker proteins, glass cover slips were coated. They were washed twice with PBS (Biochrom, Berlin, Germany). After washing with sterile A. dest., the glass cover slips were placed individually in sterile petri dishes and covered with poly-L-lysin solution (1:100 in PBS; both Biochrom) for 20 min at RT. Subsequently, the coated glass cover slips were rinsed with sterile A. dest. and dried at RT for 4 h. Until further use they were stored at 4 °C.

For vitality assay and immunofluorescence analysis ACL ligamentocytes were seeded on the coated glass cover slips with an initial density of 1 × 10^4^ cells/cm² and allowed to adhere for another 24, 48, and 72 h. Before respective staining procedures, medium was removed and cells washed in PBS. For vitality analysis the glass cover slips were immediately transferred to Live/Dead assay, while for performing the immunofluorescence analysis or histological staining, cells were fixed in 4% paraformaldehyde (PFA, Santa Cruz Biotechnology Inc., Texas, USA) for 15 min and stored at 4 °C immersed in PBS.

### 4.6. Histological Staining

ACL tissue samples (length: ~2 mm) were fixed overnight in 4% PFA solution, dehydrated and finally embedded in paraffin. After slicing, the 7 µm sections were rehydrated in a descending alcohol series and prepared for hematoxylin eosin (HE) and alcian blue (AB) staining.

For HE staining paraffin sections and glass cover slips were incubated for 6 min in Harry’s hematoxylin (Sigma-Aldrich, Munich, Germany), before rinsed in tap water and counterstained for 4 min in eosin (Carl Roth GmbH, Karlsruhe, Germany).

For AB performance, the tissue sections were rehydrated in a descending alcohol series and subsequently incubated for 3 min in 1% acetic acid before incubated for 30 min in 1% AB staining solution (Carl Roth GmbH, Karlsruhe, Germany). After rinsing in 3% acetic acid and a 2 min washing step in A. dest., ligamentocyte cell nuclei were counterstained for 5 min in nuclear fast red aluminium sulphate solution (Carl Roth GmbH).

HE and AB stained sections were covered with Entellan (Merck-Millipore, Darmstadt, Germany). Photos were taken using a DM1000 LED light microscope (Leica, Wetzlar, Germany).

### 4.7. Live/Dead Assay

Cell vitality of ligamentocytes cultured on glass cover slips, spheroids and scaffold cultures was visualized using a Live/Dead assay based on fluorescein diacetate (FDA, Sigma-Aldrich) and propidium iodide (PI, Carl Roth GmbH).

Samples were incubated for 30 s in FDA/PI staining solution (15 µg/mL FDA and 10 µg/mL PI dissolved in PBS. The green (living cells, FDA) or red (dead cells, PI) fluorescence was monitored using a SPE-II confocal laser scanning microscope (Leica, Wetzlar, Germany). Image J was used for calculation of numbers of viable and death cells.

### 4.8. Immunofluorescence Analysis of Marker Expression

The protein expression profile was assessed using confocal laser scanning microscopy. Seeded glass cover slips and ACL tissue paraffin sections, fixed in 4% PFA, were washed with Tris buffered saline (TBS: 0.05 M Tris, 0.015 M NaCl, pH 7.6), before being incubated with protease-free donkey serum (5% diluted in TBS with 0.1% Triton X100) for cell permeabilization for 20 min at RT. Subsequently, samples were incubated with primary antibodies (listed in Table 1) overnight at 4 °C in a humidified chamber. Samples were rinsed with TBS prior to incubation with donkey-anti-goat or anti-rabbit-Alexa-488 (Invitrogen, CA, USA) or donkey-anti-mouse or goat-Cy3 (Invitrogen, Carlsbad, USA) coupled secondary antibodies (diluted 1:200 in TBS with 0.1% Triton ×100 and 5% donkey serum), respectively, for 1 h at RT in a humidified chamber. Cell nuclei were counterstained using 4′,6′-diamidino-2-phenylindol (DAPI, Roche, Mannheim, Germany) and phalloidin-488 (1:100, Santa Cruz Biotechnologies) to depict F-actin cytoskeletal actin architecture. Labelled cells were washed several times with TBS, before mounting with Fluoromount mounting medium (Southern Biotech, Biozol Diagnostica, Eching, Germany) and examined by using confocal laser scanning microscopy (Leica).

### 4.9. Measurement of Cell Nuclei Diameters and Spheroid Diameters

Diameter of cell nuclei and spheroids was measured using the confocal laser scanning microscope and DAPI stained monolayer cells or FDA/PI stained spheroids after 48 h of culturing. The nuclear area was measured using ImageJ. The number of cell nucleoli per nucleus was calculated based on HE staining.

### 4.10. Quantification of Cell’s DNA Content by CyQuant Assay

The DNA content of the SV40 transfected and non-transfected control ACL ligamentocyte monolayer cultures (5263.16 cells/cm^2^) was determined using the CyQuant NF Proliferation Kit (Invitrogen, Eugene, OR, USA) according to the user manual. Calf thymus DNA served as a standard (Invitrogen). The DNA content was measured in monolayer culture at 24, 48, and 72 h in both cell populations. After rinsing with PBS (without Ca^2+^, Mg^2+^), cells were detached using 0.05% trypsin/0.02% EDTA. Subsequently, the cell pellets were washed 2× with PBS (with Ca^2+^, Mg^2+^) and stored at −20 °C until further use. The specimens were thawed at 4 °C and digested with a proteinase K solution (Sigma-Aldrich, 10 mg/mL solved in 50 mM Tris/HCl, 1 mM EDTA, 0.5% Tween20, pH 8.5) for 16 h at 55 °C. Thirty minutes of centrifugation at 10,000× *g* followed and the supernatant was stored at 4 °C. Based on the assumption that each cell contains roughly 7.7 pg DNA [34], the semi-quantitative cell nuclear content of the samples was estimated.

### 4.11. Gene Expression Analysis Using RTD-PCR

10,000 cells / cm^2^ were cultivated for 48 h in a T25 culture flask. RNA was isolated by means of the RNeasy Mini Kit (Qiagen AG, Hilden, Germany) according to the manufacturer’s protocol. Quantity and purity (e.g., 260/280 absorbance ratio) of the RNA samples were measured using the Nanodrop ND-1000 spectrophotometer (Peqlab Biotechnologie GmbH, Erlangen, Germany).

Real-time PCR analysis was carried out to compare gene expression of non-transfected and SV40 transfected cultures for several genes associated with the normal ligamentous phenotype. For cDNA synthesis 1000 ng of total RNA were reverse transcribed using the QuantiTect Reverse Transcription Kit (Qiagen AG) according to the supplier manual. 20 ng cDNA were used for each quantitative real-time PCR (qRT-PCR) reaction using TaqMan Gene Expression Assays (Life Technologies) with primer pairs and probe for scleraxis and collagen type I *versus* hypoxanthine-guanine phosphoribosyltransferase (HPRT) as reference gene (Table 2). qRT-PCR was performed using the real time PCR detector StepOnePlus (Applied Bioscience [ABI], Foster city, CA, USA) thermocycler with the program StepOnePlus software 2.3 (ABI).

The relative gene expression of the gene of interests was normalized to the HPRT expression and calculated for each sample using the ΔΔCT method as described by Livak and Schmittgen [35].

### 4.12. Statistical Analysis

Data were expressed as mean values with standard deviation. Statistics were compiled using Graphpad Prism 6 (version 6.02, GraphPad Software, USA). Statistical significances were set at a *p* value of ≤0.05. The ROUT method was used to identify outliers. Data were tested for normal distribution using the Kolmogorov-Smirnov test (α = 0.05). For analysis of normally distributed data: The unpaired two-sided t-test with Welch’s correction and the one-way ANOVA with subsequent Holm-Sidak adjustment were used e.g., for the PCR analysis. For data which did not follow Gauss distribution a one way ANOVA and Dunn’s Multiple Comparison were applied.

## 5. Conclusions

The present study showed that SV40 transfected ligamentocytes expressed typical tendon components indicating maintenance of their phenotype and sharing crucial properties with their untreated counterparts of the same donor as well as the original donor tissue used for cell isolation. This justifies them as a usable human cell culture model. However, some differences could be demonstrated such as distinct cytoskeletal alterations and a limited survival in long-term 3D cultures. Therefore, we conclude that TE with SV40 transfected cells is only possible for limited culturing periods.

In addition, we could nicely show that primary human ACL ligamentocytes can be long-term cultured on embroidered PLA and P(LA-CL)/PLA scaffolds functionalized with collagen foam and fluorination using a spheroid based seeding strategy.

## Figures and Tables

**Figure 1 ijms-21-00593-f001:**
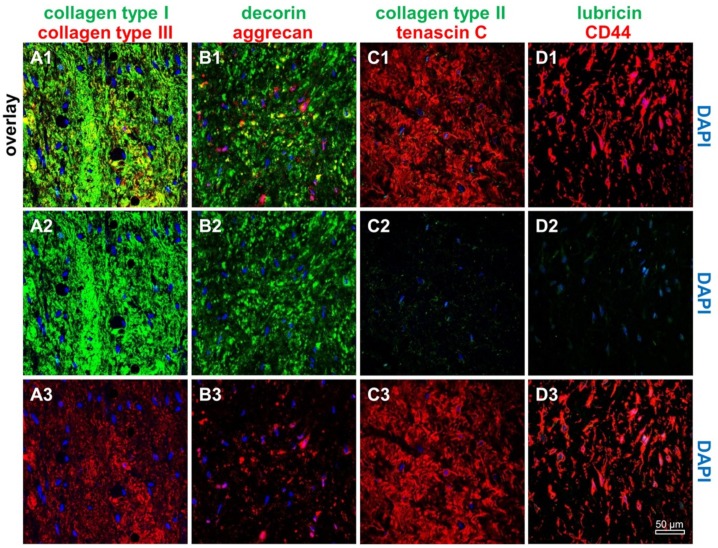
Immunohistological characterization of the human anterior cruciate ligament (ACL) donor tissue. Immunolabeling of ACL extracellular matrix components and CD44 (hyaluronan receptor) was performed using the human ACL donor tissue. (**A1**–**A3**) Collagen type I (green) and type III (red), (**B1**–**B3**) decorin (green) and aggrecan (red), (**C1**–**C3**) collagen type II (green) and the fibroblast/ligamentocyte marker tenascin C (red), (**D1**–**D3**) lubricin (green) and CD44 (red). Cell nuclei were counterstained using 4′,6′-diamidino-2-phenylindol (DAPI, blue). Scale bar: 50 µm.

**Figure 2 ijms-21-00593-f002:**
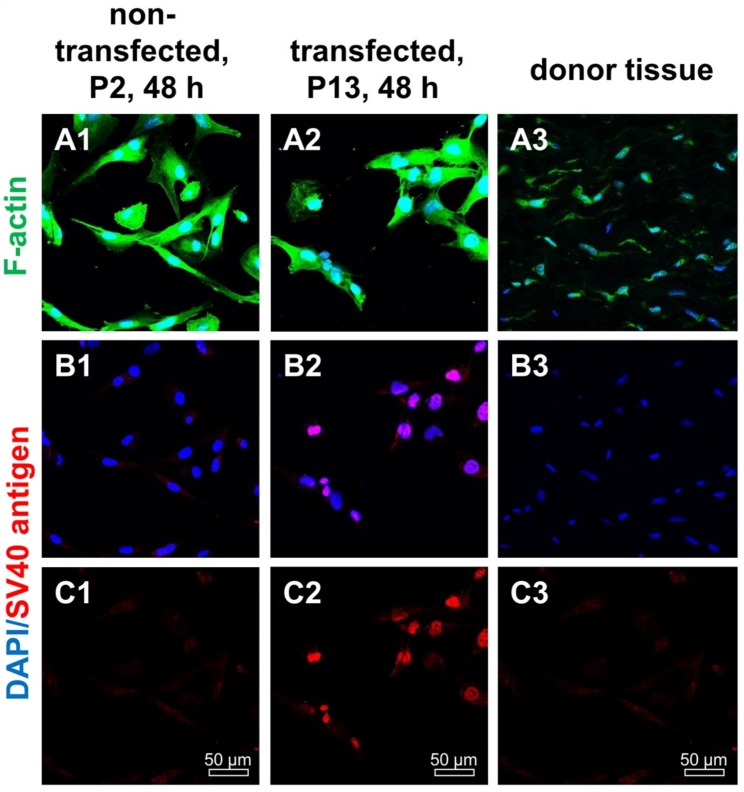
F-actin and SV40 T antigen expression in non-transfected and transfected human ACL ligamentocytes after 48 h in 2D culture as well as native tissue. (**A1**–**C1**) Non-transfected ligamentocytes, passage (P) 2, (**A2**–**C2**) ligamentocytes transfected with the SV40 plasmid (P13) and the original donor tissue in situ (**A3**–**C3**). (**A1**–**A3**) Cells were stained with phalloidin-Alexa488 (green) to depict the F-actin cytoskeleton. (**B1**–**B3**) Cells were immunolabeled with a specific anti-SV40 antibody (red), the cell nuclei were counterstained with 4′,6′-diamidino-2-phenylindol (DAPI, blue) (**A1**–**B3**). Scale bars: 50 µm.

**Figure 3 ijms-21-00593-f003:**
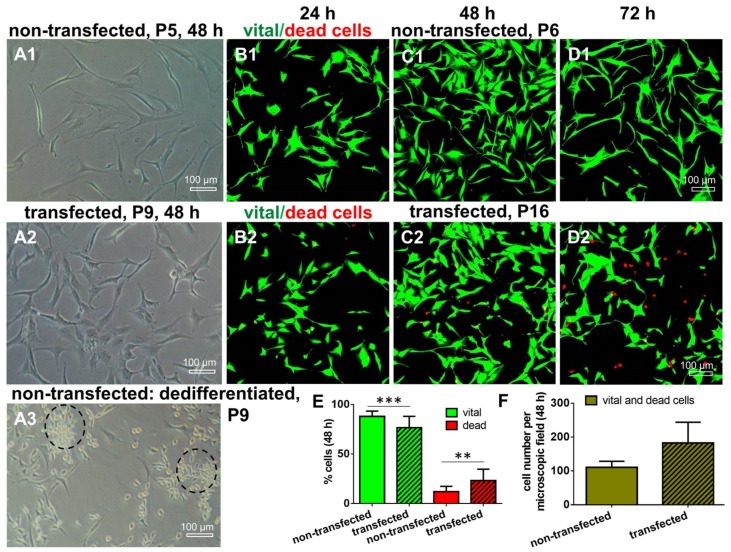
Survival of non-transfected and transfected human ACL ligamentocytes. (**A1**,**A2**) Images of non-transfected and transfected cells. (**B1**,**C1**,**D1**) Live/Dead assay of non-transfected cells and of SV40 transfected cells cultured in monolayer for 24, 48, and 72 h (**B2**,**C2**,**D2**). (**A3**) Non-transfected cells dedifferentiated in monolayer passage (P9) reflecting clusters of round proliferating cells (dashed circles). (**E**) Comparison of the percentage of vital (green) and dead (red) cells in non-transfected and SV40 transfected ACL ligamentocytes (48 h). (**F**) Comparison of the total number of cells per microscopic field after 48 h monolayer culture of non-transfected and transfected cells (n = 3, with three evaluated microscopic fields each). Scale bars: 100 µm. * *p* < 0.05, *** *p* < 0.001.

**Figure 4 ijms-21-00593-f004:**
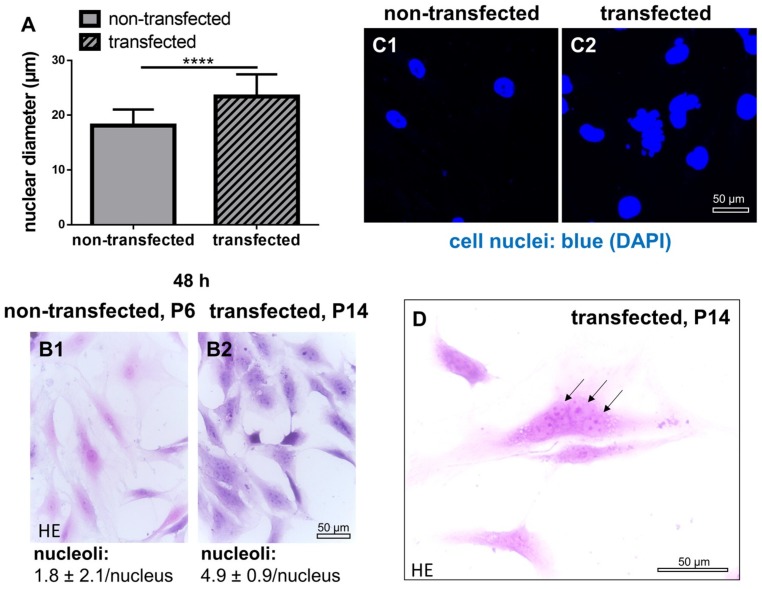
Size and shape of cell nuclei, numbers of nucleoli of non-transfected and SV40 transfected ACL ligamentocytes in 2D culture. (**A**) Nuclear diameter. (**B1**,**B2**) Number of nucleoli in non-transfected (B1) and transfected (B2) ACL ligamentocytes stained using hematoxylin eosin (HE). Nuclear morphology in non-transfected (**C1**) and transfected cells (**C2**). Cell nuclei of ACL ligamentocytes were visualized using 4′,6′-diamidino-2-phenylindol (DAPI, blue). (**D**) Random cell division without cytokinesis was shown by HE staining. Arrows: Multiple cell nuclei within one singular cell. Scale bars: 50 µm. **** *p* < 0.0001.

**Figure 5 ijms-21-00593-f005:**
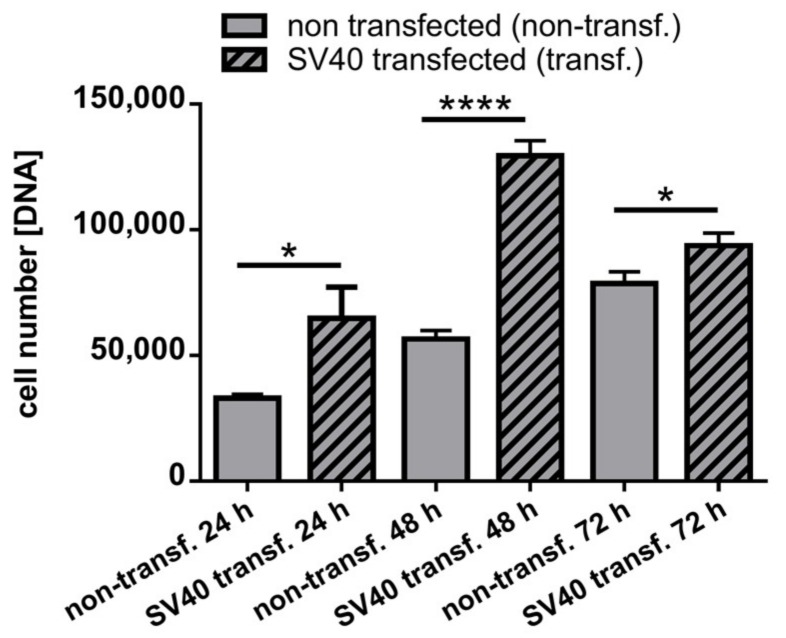
DNA content of 2D cultured non-transfected and SV40 transfected ACL ligamentocytes. Non-transfected and transfected cells were cultured for 24, 48, and 72 h and DNA content was determined by CyQuant assay. Calf thymus DNA served as a standard. Three independent experiments were performed. Transf.: transfected, * *p* < 0.05, **** *p* < 0.0001.

**Figure 6 ijms-21-00593-f006:**
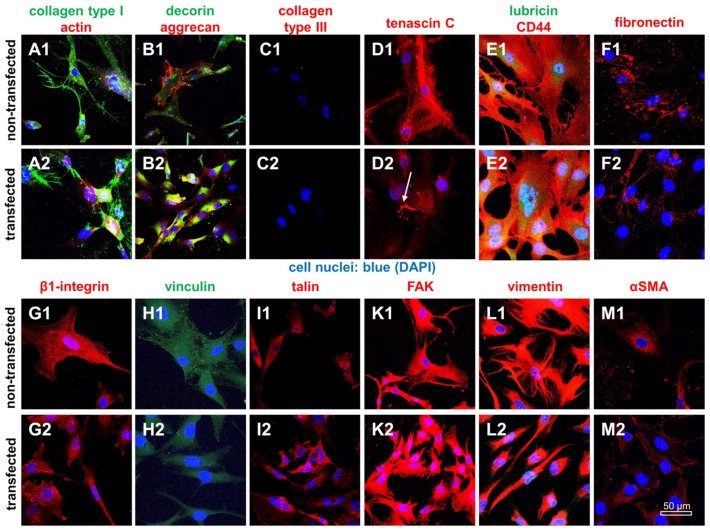
Expression profile of ligament ECM components and of cytoskeletal constituents in 2D cultured non-transfected and SV40 transfected ACL ligamentocytes. Collagen type I (green), actin (red) (**A**); decorin (green), aggrecan (red) (**B**); collagen type III (red) (**C**); tenascin C (red) (**D**); lubricin (green), CD44 (red) (**E**); fibronectin (red) (**F**); β1 integrin (red) (**G**); vinculin (green) (**H**); talin (red) (**I**); focal adhesion kinase (FAK) (red) (**K**); vimentin (red) (**L**); and α-smooth muscle actin (αSMA) (red) (**M**). Arrow: Extracellular deposition of tenascin C. (**A1**–**M1**) Non-transfected. (**A2**–**M2**) Transfected ligamentocytes. Cell nuclei were counterstained using 4′,6′-diamidino-2-phenylindol (DAPI, blue). Scale bars: 50 µm.

**Figure 7 ijms-21-00593-f007:**
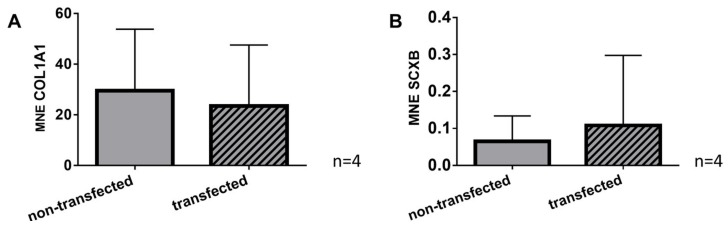
Gene expression profile of ligament ECM components in 2D cultured non-transfected and SV40 transfected ACL ligamentocytes. Gene expression of collagen type I (COL1A1) (**A**) and scleraxis (SCXB) (**B**). MNE: Mean normalized expression. Four independent experiments were performed.

**Figure 8 ijms-21-00593-f008:**
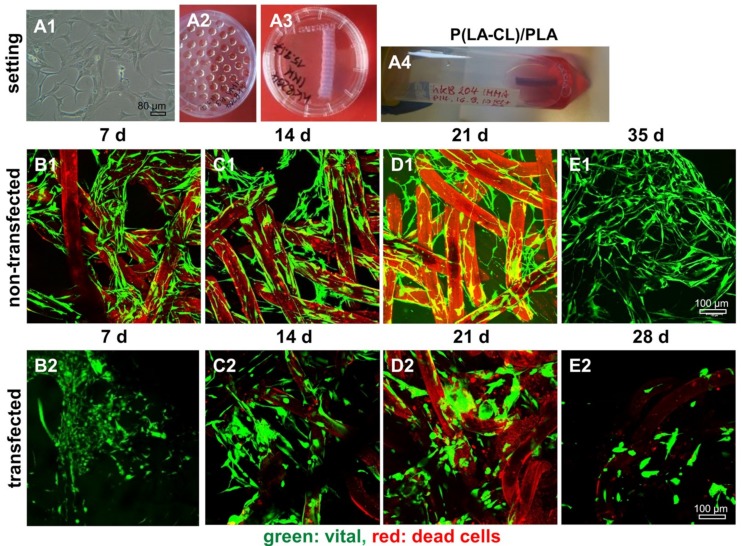
Representative microscopic fields of non-transfected and SV40 transfected ACL ligamentocytes on P(LA-CL)/PLA scaffolds. Experimental design, cell expansion (**A1**); self-assembly using hanging drop method (50,000 cells per spheroid) (**A2**); macroscopic image of wet P(LA-CL)/PLA scaffolds (**A3**); and dynamical rotatory scaffold culture (**A4**). (**B1**–**E2**) Live/Dead staining of cultures maintained up to day 35 (non-transfected cells, (**B1–E1**)) and day 28 (SV40 transfected cells, (**B2–D2**)). Vital cells: Green. Dead cell: Red. Scaffold fibers: Slightly red colored. Three independent experiments were performed. Scale bars: 100 µm.

**Figure 9 ijms-21-00593-f009:**
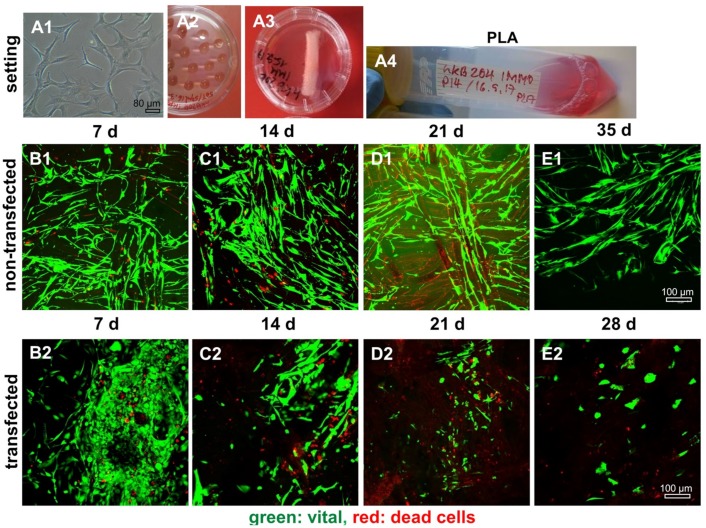
Representative microscopic fields of non-transfected and SV40 transfected ACL ligamentocytes on PLA scaffolds. Experimental design, cell expansion (**A1**), self-assembly using hanging drop method (50,000 cells per spheroid) (**A2**), macroscopic image of the wet PLA scaffolds (**A3**), and dynamical rotatory scaffold culture (**A4**). (**B1**–**E2**) Live/Dead staining of cultures up to day 35 (non-transfected) and day 28 (SV40 transfected) cells. (**B1–E1**) Non-transfected. (**B2–E2**) SV40 transfected ligamentocytes. Vital cells: Green. Dead cells: Red. Scaffold fibers: Slightly red colored. Three independent experiments were performed. Scale bars: 100 µm.

**Figure 10 ijms-21-00593-f010:**
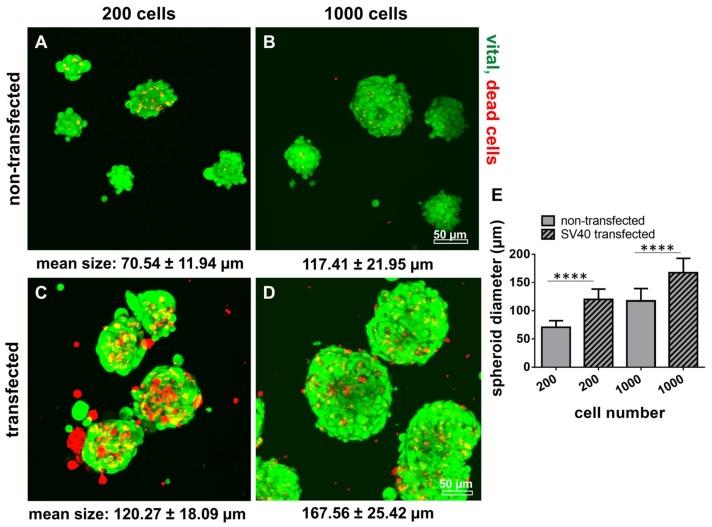
Microspheroid formation by non-transfected and SV40 transfected human ACL ligamentocytes: Spheroid size as well as cell vitality. 200 (**A**,**C**) and 1000 (**B**,**D**) human ACL ligamentocytes per spheroid either non-transfected (A,B) or SV40 transfected (**C,D**) were allowed to aggregate for 5 d using a spheroid multiwell plate. Live/Dead staining was performed and spheroid diameters were measured (**E**, 48 h). Scale bars: 50 µm. **** *p* < 0.0001.

**Table 1 ijms-21-00593-t001:** Antibodies used for immunolabeling.

Target	Primary Antibody	Dilution	Secondary Antibody	Dilution
αSMA	mouse-anti-human, Sigma-Aldrich	1:50	donkey-anti-mouse cy-3, Invitrogen	1:200
β1-integrin	mouse-anti-human, Merck-Millipore, Darmstadt, Germany	1:30	donkey-anti-mouse cy-3, Invitrogen	1:200
aggrecan	mouse anti human R&D systems, Minneapolis, USA	1:30	donkey-anti-mouse cy-3, Invitrogen	1:200
CD44	mouse-anti-human, Cell signalling Technology, Danvers, USA	1:50	donkey-anti-mouse cy-3, Invitrogen	1:200
collagen type I	goat anti human, Abcam, Cambridge, UK	1:50	donkey anti goat, Alexa Fluor 488, Invitrogen, Carlsbad, USA	1:200
collagen type II	rabbit anti human, Acris Laboratories, Hiddenhausen, Germany	1:50	donkey anti rabbit, Alexa Fluor 488, Invitrogen	1:200
collagen type III	mouse anti human Acris Laboratories	1:30	donkey-anti-mouse cyanine-3 (cy3), Invitrogen	1:200
decorin	rabbit anti human, Acris Laboratories	1:30	donkey anti rabbit, Alexa Fluor 488, Invitrogen	1:200
elastin	mouse anti human Acris Laboratories	1:30	donkey-anti-mouse cy-3, Invitrogen	1:200
fibronectin	mouse-anti-human, Dianova, Hamburg, Germany	1:30	donkey-anti-mouse cy-3, Invitrogen	1:200
focal adhesion kinase	mouse-anti-human, BD Transduction Laboratories, Ca, San Jose, USA	1:30	donkey-anti-mouse cy-3, Invitrogen	1:200
lubricin	rabbit-anti-human, Abcam, Cambridge, UK	1:30	donkey anti rabbit, Alexa Fluor 488, Invitrogen	1:200
mohawk	rabbit-anti-human, Biozol, Eching, Germany	1:30	donkey anti rabbit, Alexa Fluor 488, Invitrogen	1:200
SV40 T antigen	mouse-anti-human, Merck-Millipore	1:30	donkey-anti-mouse cy-3, Invitrogen	1:200
talin	mouse-anti-human, Sigma-Aldrich	1:30	donkey-anti-mouse cy-3, Invitrogen	1:200
tenascin C	mouse-anti-human, GeneTex Inc. Biozol	1:30	donkey-anti-mouse cy-3, Invitrogen	1:200
VEGF	mouse-anti-human, R&D Systems	1:30	donkey-anti-mouse cy-3, Invitrogen	1:200
vimentin	mouse-anti-human, Dako Cytomation, Hamburg, Germany	1:50	donkey-anti-mouse cy-3, Invitrogen	1:200
vinculin	mouse-anti-human 1:50, Sigma-Aldrich	1:50	donkey-anti-mouse cy-3, Invitrogen	1:200

**Table 2 ijms-21-00593-t002:** Primer sequences used for RTD-PCR.

Gene Symbol	Species	Gene Name	Amplicon Length (bp *)	Assay ID
*COL1A1*	Homo sapiens	collagen type I, alpha1 chain	66	Hs00164004_m1
*SCXB*	Homo sapiens	scleraxis homolog B	63	Hs03054634_g1
*HPRT*	Homo sapiens	hypoxanthine-guanine phosphoribosyltransferase	100	Hs99999909_m1

* base pairs.

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
