# Peer review of "SV40 Transfected Human Anterior Cruciate Ligament Derived Ligamentocytes—Suitable as a Human in Vitro Model for Ligament Reconstruction?"

_ijms, 2020, doi:10.3390/ijms21020593_

Round 1

Reviewer 1 Report

Manuscript ID: ijms-691728

Title: SV40 transfected human anterior cruciate ligament derived ligamentocytes - suitable as a human in vitro model for ligament reconstruction?

Reviewer comments

No line numbering through the manuscript. Overall it is a well presented and executed study, which covers an important topic of the utilization of immortalized human cell lines in tissue engineering applications. Authors present a compelling set of experimental evidence that ligamentocytes can be immortalized using transfection with SV40 vector, which is of potentially high importance for similar work with scarcely available human cell lines. There are only minor corrections per specific comments section below.

Specific comments

Page 2, line 5: “In contrast, animal derived cells usually show inter-species differences scrutinizing transferability of results. The advantages of immortalized cell lines are lacking or delayed senescence and more stable maintenance of phenotype than in non-immortalized cells.” This paragraph represents an example of a messy and confusing style. What are you trying to convey in these sentences?

Page 3, line 1: “Graphical abstract: Cell isolation and culture procedures.” The HE and AB acronyms are not explained. Best is to write down the expanded versions for both acronyms.

Page 6, line 1: “A1, A2: Invert microscopical images of…” It doesn’t matter what kind of technique was used to acquire these images. The phrasing used makes it sound nonsensical.

Author Response

Dear Editor,                                                                                                   10th January 2020

The authors would like to thank the reviewer for carefully reading the text and their constructive encouraging comments. We modified the manuscript according to their suggestions and comments. A list of changes is reported below. All changes performed are indicated in red and underlined in the revised version of the manuscript. The manuscript has been proof read for English style corrections. We decided also to insert a panel including mohawk immunolabeling (mentioned before as „not shown“) as supplemental figure 1. Please refer to our point by point reply below.

We hope you will find this manuscript suitable for publication in “IJMSnow. Please do not hesitate to contact me anytime for questions regarding this manuscript.

Sincerely,

Univ.-Prof. Dr. Gundula Schulze-Tanzil

POINT BY POINT REPLY

Reviewer 1

Reviewer comments

No line numbering through the manuscript.

Response: we added line numbering now

Overall it is a well presented and executed study, which covers an important topic of the utilization of immortalized human cell lines in tissue engineering applications. Authors present a compelling set of experimental evidence that ligamentocytes can be immortalized using transfection with SV40 vector, which is of potentially high importance for similar work with scarcely available human cell lines. There are only minor corrections per specific comments section below.

Specific comments

Page 2, line 5: “In contrast, animal derived cells usually show inter-species differences scrutinizing transferability of results. The advantages of immortalized cell lines are lacking or delayed senescence and more stable maintenance of phenotype than in non-immortalized cells.” This paragraph represents an example of a messy and confusing style. What are you trying to convey in these sentences?

Response: we rewrote this paragraph now.

Page 3, line 1: “Graphical abstract: Cell isolation and culture procedures.” The HE and AB acronyms are not explained. Best is to write down the expanded versions for both acronyms.

Response: we wrote the complete version now.

Page 6, line 1: “A1, A2: Invert microscopical images of…” It doesn’t matter what kind of technique was used to acquire these images. The phrasing used makes it sound nonsensical.

Response: we deleted „invert microscopical“ (legend of figure 3).

Reviewer 2 Report

The article submitted by Gundula Schulze-Tanziland co-workers, entitled “SV40 transfected human anterior cruciate ligament derived ligamentocytes - suitable as a human in vitro model for ligament reconstruction?” presents a set of very interesting data introducing potential new experimental model for tissue engineering studies. The study overall is coherent and comprehensive.

- The introduction is concise and well brings to the topic of the paper indicating the identified differences in the cell proliferation and metabolism due to immortalization process. To figure out if the immortalized ACL ligamentocytes might be a good cell-based engineering model the Authors performed a set of experiments comparing immortalized cells with the primary, also analyzing their ability for colonizing different embroidered scaffold compositions.

- The methods for validation the hypothesis are properly chosen and described in sufficient details.

- The studies performed, are properly designed and controlled, replicated suitable. Statistical analysis of data is performed properly. The Authors use of immunofluorescence staining followed by microscopic analysis for identification of ECM components, cytoskeleton constituents and cell size differences which is more appropriate for adherent cells than flow cytometry as indicated in discussion.

- The presented data are of high quality.

- The conclusions reached, are consistent with the presented data.

I do recommend the publication of the paper as suitable for International Journal of Molecular Sciences.

Author Response

Reviewer 2

The article submitted by Gundula Schulze-Tanziland co-workers, entitled “SV40 transfected human anterior cruciate ligament derived ligamentocytes - suitable as a human in vitro model for ligament reconstruction?” presents a set of very interesting data introducing potential new experimental model for tissue engineering studies. The study overall is coherent and comprehensive.

- The introduction is concise and well brings to the topic of the paper indicating the identified differences in the cell proliferation and metabolism due to immortalization process. To figure out if the immortalized ACL ligamentocytes might be a good cell-based engineering model the Authors performed a set of experiments comparing immortalized cells with the primary, also analyzing their ability for colonizing different embroidered scaffold compositions.

- The methods for validation the hypothesis are properly chosen and described in sufficient details.

- The studies performed, are properly designed and controlled, replicated suitable. Statistical analysis of data is performed properly. The Authors use of immunofluorescence staining followed by microscopic analysis for identification of ECM components, cytoskeleton constituents and cell size differences which is more appropriate for adherent cells than flow cytometry as indicated in discussion.

- The presented data are of high quality.

- The conclusions reached, are consistent with the presented data.

Response: we thank the reviewer for his/her encouraging comments.
